# Minjung Theology of Korea and Ecological Thinking: Focusing on the Theological Imagination of Ahn Byung-Mu

**Jongman Kim [1] and Andrew Eungi Kim [2,\*]**

1   Institute for Religion and Civic Culture, Kyung Hee University, Seoul 02447, Republic of Korea; kjmif@khu.ac.kr

2   College of International Studies, Korea University, Seoul 02841, Republic of Korea

\*   Correspondence: aekim@korea.ac.kr

**Abstract:** Environmental, social, and governance (ESG) criteria, currently used as a set of standards by socially conscious investors to evaluate a company's operations before investing, are becoming an important global trend today. In particular, environmental and ecological crises are increasingly being seen as issues that will determine the sustainability of human civilization. Scholars of religion have been paying more attention to the issue as well. In fact, religion and environmentalism have emerged as sub-disciplines in, among others, religious ethics, religious studies, the sociology of religion, and theology. In view of this development, this paper aims to reexamine Minjung theology, literally meaning "the people's theology", which arose as a form of liberation theology in South Korea in the 1970s, from an ecological perspective, particularly focusing on the former's view on the relationship and interrelationship between the individual and the environment. The paper pays special attention to the work of Ahn Byung-Mu, a founding scholar of Minjung theology, shedding light on the connection between his concept of *gong*, literally meaning "publicness", and ecology, the characteristics of his ecological thoughts and their relevance to his view of god, and his views on *bapsanggongdongche*, literally meaning "the table community".

**Keywords:** liberation theology; Ahn Byeong-Mu; ecology; *gong*; publicness; *bapsanggongdongche*; the table community; sharing

## 1. Introduction

There has been growing scholarly attention being paid to the relationship between religion and ecology. Scholars and practitioners alike of various religious traditions are increasingly engaging in ecological debates, hoping to show that their religions are relevant to perhaps the most pressing issue facing humanity today. Due to such efforts, there is an increasing awareness that we must broaden our understanding of our relationships with the planet in order to think ecologically. Realizing how we are interconnected with and dependent on the environment is at the core of ecological thinking. Ecology demonstrates how ecosystems are relational systems in which humans are interconnected with other biological systems, such as plants and animals.

In Christianity, for example, major Christian denominations are altering their notions of stewardship, the common good, and sin, among others, to endorse the biblical calling for the protection and guardianship of God's creations and human responsibility for their care (Wikipedia 2023; see Bradley 2020; Schlogl-Flierl 2023). There are supposedly many verses in the Bible that mention the protection of the environment, implying that God has tasked humans with responsibility for taking care of what God created. Accordingly, "green Christianity", which is non-denominational or pan-denominational Christian environmental activism that promotes environmental awareness and action at the church, community, and national levels, emphasizes the biblical or theological basis for safeguarding the environment (Nita 2016; Wallace 2010).[1] Green Christians note that the Bible places

a strong emphasis on stewardship rather than ownership, saying that the earth is still the Lord's (Psalms 24:1) and does not belong to the people who live on it (Wikipedia 2023). According to Leviticus 25:23, "The land must not be sold permanently, because the land is mine and you reside in my land as foreigners and strangers." Christian environmentalists oppose actions and policies that endanger the planet's survival or health because they adhere to the stewardship philosophy. They are particularly concerned with pollution, habitat destruction, and the extensive reliance on non-renewable resources, which either negatively impact climate change or jeopardize the wellbeing of the ecosystem.

Other world religions have also emphasized their concern for the environment. Buddhism in recent years has developed a reputation as an "environmental religion", as many Buddhists draw from the teachings of Buddha to inform their engagement with the environment and activism toward climate change (see Tucker and Williams 1997; Batchelor and Brown 1992; Payne 2010). While Buddha himself did not give specific teachings on environmentalism, as it was not an issue during his lifetime, he did set rules for monks and nuns as well as his followers, which has served as a guideline for environmentally friendly behaviors. For example, recognizing that the behavior of his followers can affect the wellbeing of local communities, Buddha instructed monks and nuns never to relieve themselves in or near areas where people bathe or drink the water (BBC 2023). He also taught them not to kill other living creatures and that they should not disturb the established habitat of other creatures, especially when building new neighborhoods. Hinduism holds that everything in the cosmos is sacred and divine because it is a manifestation of Brahman, the ultimate reality (Riya 2023). Hindus see nature as a manifestation of the divine that should be revered and protected as a result of this belief. The teachings of Hinduism place a strong emphasis on the idea that everything in the cosmos is interconnected and that every element has an impact on everything else (Riya 2023; see Chapple and Tucker 2000; Coward 1997; Framarin 2012). In this way, Hinduism has a strong connection to the natural world. Islamic teachings place a strong emphasis on protecting the environment (Facts about Islam 2017; see Foltz 2006; Haq 2003). Muslims believe in God's unconditional sovereignty over nature and stress human beings' role as God's agent in protecting and preserving the environment, as the Qur'an designates humankind as stewards of the earth. Islam also teaches people to refrain from being wasteful and wasting natural resources. While not amounting to championing animal rights, Islam also views that each animal species has a part to play in preserving harmony and balance on earth (Facts about Islam 2017).

Given the fact that many religious traditions have taken up the issue of environmentalism, this paper wishes to probe the view of liberation theology, arguably one of the most challenging theological developments in the past half century, on ecological issues. More specifically, the paper attempts to examine the approach of Korean liberation theology, namely Minjung theology, to ecological issues. Before turning to the latter, it would be pertinent to mention a few words about "ecological liberation theology." In a nutshell, an underlying belief of ecological liberation theology is that theology should be committed not only to poverty alleviation but also to protecting and preserving the environment. Ecological liberation theology views "the relationship between poverty, ecological devastation, and oppression as an interrelated structural problem" (Holden et al. 2017, p. 1). The need for an ecological liberation theology has been prompted by neoliberalism, which has engendered "a local environment of unregulated, unplanned development without implementing safeguards for the protection of poor rural and urban communities and the natural environment" (Holden et al. 2017, p. 1; see Castillo 2019; Scharper 2006). In seeing the connection between oppression and poverty as well as how this has had a direct influence on the environment, ecological liberation theology sees modernization processes from the perspective of the impoverished and those who have been victimized by state-led development and, more recently, by human-made and natural disasters caused by climate change (Holden et al. 2017, p. 36). Ecological liberation theologians urge everyone to rethink about humanity and nature, arguing that we need to constantly remind ourselves about the interconnectedness of human beings and nature and to live in harmony with nature.

Leonardo Boff, a founding figure of the liberation theology movement who is perhaps the most renowned liberation theologian to systematically address ecological issues, integrates the ecological problem into the larger problems affecting the modern world and the latter's view of development, posing the following question:

> What is the fate and future of planet Earth if we prolong the logic of plunder to which our development and consumer model have accustomed us? What can the poor two-thirds of humankind hope for from the world? (Boff 1995, p. 75)

For Boff, ecology, be it in Latin America and elsewhere, "must be a social ecology, one that takes social and economic questions of poverty and injustice seriously and moves from a logic of profit to one of the common good, not only for humans but for all creatures" (Scharper 2006, p. 60; see Scharper 1997; Boff 1997; Gebara 1999). Boff thus seeks to construct an "alternate and integral modernity", one that recognizes the significant advances of modernity with regard to democratization and improvement of human rights but rejects the ideologies of exploitation and devastation of the environment (Scharper 2006, p. 60). Also, in suggesting that ecology should be an additional "pillar" of liberation theology, he sees the connection between ecology and social justice as follows:

> The core of liberation theology is the empowerment of the poor to end poverty and achieve the freedom to live a good life. In the 1980s, we realized that the logic supporting the exploitation of workers was the same as that supporting the exploitation of the earth. Out of this insight, a vigorous liberation eco-theology was born. To make this movement effective, it is important to create a new paradigm rooted in cosmology, biology, and complexity theory. A global vision of reality must always be open to creating new forms of order within which human life can evolve. The vision of James Lovelock and V. I. Vernadsky helped us see not only that life exists on Earth but also that Earth itself is a living organism. The human being is the highest expression of Earth's creation by virtue of our capacity to feel, think, love, and worship. (Boff 2016)

In view of this preliminary observation on liberation theology's view on environmental issues, the paper turns to the question of Minjung theology's approach to ecology. It must be admitted at the outset that, apart from the above description of ecological liberation theology, there is no unique view of Korean liberation theology on ecological issues. Also, there has been only scant attention paid to the link between Minjung theology and ecology (see Kwon 2012; H. Lee 2002; Chang 2015). However, one issue that does deserve special attention is the place of nature or ecology in the theological thoughts of Ahn Byeong-Mu (1922–1996), a founding figure of Mnjung theology in South Korea (henceforth Korea).[2] Although Ahn himself never wrote any work specifically focused on the theme of "environment" or "ecology", fragmentary commentaries appear in some of his writings. In the late 1990s, in particular, when the seriousness of ecological destruction emerged, Ahn did speak of the need for and importance of the coexistence between humans and nature (see H. Lee 2002, p. 279).[3] In addition, Ahn's perception of ecology is revealed in a conversation he had with a *Christian Thought* reporter on June 20, 1996, in which he expressed his conviction that the cause of today's ecological crisis is the Western Christian doctrine that targets nature for conquest. Such a view is in parallel with, for instance, the perspective of Lynn White, Jr., who wrote in his article entitled "The Historical Roots of Our Ecological Crisis" (White 1967) that the cause of the ecological crisis facing humanity lies in Western Christianity, and with John Passmore (1974), who discusses Christian responsibility for the exploitation of nature and ecological crisis in his book *Man's Responsibility for Nature: Ecological Problems and Western Tradition* (see J. Kim 2020). Ahn says that the ecological crisis can be overcome by expanding the perspective of life by reinterpreting the Holy Spirit with the Eastern concept of *qi*, literally meaning "life force" or "energy", which is similar to the Greek word "pneuma" and the Hebrew word "ruach."

Until now, Ahn's Minjung theology has received considerable attention from scholars in and outside of Korea, including Jürgen Moltmann. However, one topic that has been

wholly neglected is Ahn's theological thought related to ecology, as there has not been even a single article directly related to the issue. In order to redress this imbalance, this paper examines Ahn's theological thoughts from the perspective of "ecology." In particular, the study aims to reconstruct his ecological thoughts by exploring his philosophy of *gong* (公), literally meaning "publicness", which forms the core of the ethical scale of Ahn's Minjung theology (Choi 2006a, 2006b). The paper also traces the origin of ecological thought in Ahn's life, followed by an analysis of his ideas behind *bapsanggongdongche* ("the table community"). The paper then examines how ecology relates to Ahn's view of God. The paper concludes by assessing the characteristics of Ahn's ecological thought. In short, although it is unfortunate that Ahn's ecological interest and awareness did not result in any published work, the significance of the study is to shed new light on his ecological thinking, which approaches the ecological crisis based on Eastern thought and understands it anew in relation to his own de-Westernized view of God.

## 2. Ahn Byeong-Mu's Theology and Ecology

### 2.1. Ecological Thoughts of Ahn Byeong-Mu

Ahn Byeong-Mu was born in 1912 in Sinanju, South Pyongan Province. However, his "hometown", i.e., a home base where he grew up, was Gando (part of northeastern China today), which was a center of Korea's independence movement against Japanese colonial rule (1910–1945). Ahn's family moved to Gando when he was not even a year old due to economic difficulties. Ahn's father earned his livelihood as a doctor of oriental medicine in the region. However, Ahn detested his father's strict Confucian, patriarchal way of thinking and manners (Ahn 1993a, pp. 15–16).[4] Could Ahn's conversion to Christianity, which happened without anyone's help or persuasion, have been due to his emotional rejection of his father? This possibility is revealed through N.-i. Kim (2007, p. 49), the author of Ahn Byeong-mu's intellectual biography:

> [Ahn's father said] "There are many great people like Confucius, but why on earth are you bringing in a Western religion to disgrace your family?" [Ahn Byeong-mu replied] "Father, I hate both Confucius and Mencius. If Confucianism means that you can just bully mother like that and that you can just drink day in and day out, then I hate all of that." At that time, the boy (Ahn Byung-Mu) really chose Christianity for himself with a very practical motive: to stop his father from drinking and womanizing.

Ahn became absorbed in the spirit of independence, self-reliance, and resistance due to the influences of teachers at Eunjin Middle School in Yongjeong, Gando, who were imbued with strong anti-Japanese nationalism through their involvement with their church. According to church historian Min Gyung-bae, Christianity in Gando, which Ahn encountered, had characteristics different from the conservative Christianity in northwestern Korea at the time. The former was characterized by nationalism and subjective spirit as it was shaped by Christians who resettled from the Ganseo and Gwanbuk districts in the northeastern part of Korea, the area that was influenced by the more liberal theology of Canadian missionaries. The latter, on the other hand, which was centered in Pyongyang, had conservative, fundamentalist, and post-historical tendencies, as its followers were largely composed of middle-class Korean Christians, and it was propagated by the missionaries from the eastern United States who held conservative theological views (Min 1978, p. 84).

From a young age, Ahn was able to develop a sense of resistance to oppression and exploitation, as Korea was under the draconian rule of Japanese authorities. Those sentiments were also reinforced by his feelings against his father's stern Confucian and patriarchal worldviews and manners. More importantly, perhaps the spirit of resistance against oppression was shaped by Ahn's admiration of the will of Jesus, who, at the risk of his own death, was on the side of the poor and the powerless to resist the religious power of Israel and the political power of Rome, both of which tried to oppress and exploit the people (Ahn 1997, pp. 323–24).

From this, we can see that Ahn's "ecological thought"—i.e., that ecology is open to all and cannot be "monopolized" by any person or group and that humans are part of the web of life, not separate from other humans and from nature—was born from his motive of resisting the centralization of power. Power is characterized by monopolization and privatization of publicness (*gong*) that must be co-owned by, and be for, all who have a stake in its maintenance. The way to maintain power is to suppress *qi*,[5] literally meaning "material energy", "life force", or "vital energy", of *minjung*, literally meaning the people. However, the people's *qi* has a completely different way of existence than those in power think. Ahn argues that "*qi* has been in existence ever since the beginning of the universe." He understands the *qi* of *minjung* by expanding it to the socio-historical level, not simply limiting it to the cosmological and ontological level.

> In the world before any ideas or concepts, *qi* has been the foundation that enables the existence of all beings. Just as *qi* forms the basis of all life, people are the basis of social history. *Qi* must always flow. In all of life, if the flow of *qi* is blocked, it is bound to get sick and die. Just as *qi* flows by itself and saves all life, so does *minjung*. When *minjung* live, the whole society lives. (M.-s. Kim 2012, p. 158)

Ahn expresses *minjung* as a descriptive adjective, not a noun, and interprets it (the people) as "as they are by themselves", as nature does, in a pristine state and untouched by artificial means. He is also opposed to the conceptualization of *minjung* because a conceptualized existence is trapped in a fixed frame, and when it does, it becomes separate from existence itself, which is a process of change. Ahn believes that when *minjung* is conceptualized, the true character of the living and dynamic people disappears, and only the abstract and "taxidermied" *minjung* remain. Therefore, Ahn understands *minjung* as living beings. Just as living organisms cannot be "labeled", conceptualizing *minjung* results in taxidermy or killing of *minjung* (M.-s. Kim 2012, pp. 155–56). M.-s. Kim (2012, p. 156) explains Ahn's understanding of *minjung* as follows:

> When we leave behind the image of *minjung*, which is a product of human ideology, and look at *minjung*, we encounter the true nature of the people. *Minjung* must be experienced not as a concept or within any concept, but within the historical field, and should be experienced within our lives and history.

In this way, just as *minjung* is not an object of perception in a state of separation between the subject and the object but an object of experience and practice, *minjung* functions as "a place of experience, not cognition" (M.-s. Kim 2012, p. 165). Such understanding of *minjung* ultimately acts as a mechanism to basically block the will of those in power to monopolize publicness (*gong*) by discouraging the spirit of the people. In short, the fact that the reality of publicness, which is not privatized by those in power, is continuously being realized on this land through *minjung* events is in line with the ecology that is open to all and is not monopolized. Therefore, it can be inferred that Ahn's ecological thoughts were born and formed from his understanding of *minjung* and the Eastern way of thinking about *qi*.

### 2.2. Bapsanggongdongche (The Table Community) as an Organic Body

At the core of Ahn's theological thought is *minjung* and, as an extension, the construction of a *minjung* community. Ahn's conceptualization of *bapsanggongdongche* ("the table community") is intimately tied to his experience as a university student and as a young adult. He took the lead in a community movement centered on Christian laity as early as his early twenties. While attending Seoul National University, he organized a group for Christians, namely Ilsinhoe, and continued to hold meetings after graduation. Shortly after, when the Korean War (1950–1953) broke out, members of Ilsinhoe went their separate ways. Following the end of the war, Ahn (1993a, pp. 24–25) called upon the members with the determination to create a new community akin to a church. At this time, Ahn planned to build a community he had dreamed of by using Hyanglinwon,[6] located in Namsan, Seoul. The new community Ahn envisioned was a church of the laity and a common eco-

nomic community. A church of the laity was a church where lay people led the liturgy and administration without an ordained pastor. It was a community where people working in various fields formed a commune to live and study like monks and pursue networks with people outside the community. It was also a common economic community based on the principle of the spirit of Jesus: "If anyone comes to me and does not hate father and mother, wife and children, brothers and sisters—yes, even their own life—such a person cannot be my disciple" (Luke 14:26). For example, they adhered to economic management rules whereby those who were financially well off paid more, and those who could not afford it paid less. They also founded a cram school, using the income from that business for common purposes. Ahn believed that the existing churches had degenerated into a marketplace by heightening religious sensibility through guilt and utilizing religious resources, which amounted to "salvific goods." Therefore, he harshly criticized the corruption of churches and church leaders and dreamed of a new Christian community (N.-i. Kim 2007, p. 96). A theologian later assessed this new lay community as "a solitary untamed cry against a corrupt church and an untamed cry for salvation for a wounded world" (J.-H. Kim 2006; cited from N.-i. Kim 2007, p. 96).

However, Ahn's attempt at building a community was halted in less than three years of its launch due to strains caused by, among others, a considerable difference in income depending on members' jobs, the lack of time due to work, and the issue of some members studying abroad. At the same time, the community gradually transformed into the form of a "church" due to increasing family relationships and the expanded influx of outsiders. When the community he had founded developed into an ordinary church, Ahn self-mockingly expressed his thoughts by saying, "I drew a cat while trying to draw a tiger" (N.-i. Kim 2007, p. 99). Nevertheless, what is noteworthy about all of this is the image of the community Ahn drew in his head. Ahn dreamed of a communal community, i.e., *minjung* of Jesus, in harmony with one another in "the table community", as in the Bible, where "All the believers are one in heart and mind. No one claimed that any of their possessions were their own, but they shared everything they had" (Acts 4:32). Such a community was a community that experienced the Kingdom of God by restoring the sharing of materials with the public. However, as described above, the community Ahn dreamed of collapsed as a kind of class system emerged between the haves and the have-nots, and the table community eventually degenerated into Holy Communion with only religious rituals remaining (Ahn 1993a, pp. 388–89). Regarding this, Ahn (1993a, p. 389) confesses his regretful feelings as follows:

> Whether it was by Paul or not, it was a great mistake for the leaders of the early church to turn the dinner they shared together into a sacrament. As a result, the table community was destroyed, and only religious rituals remained. Only the religious ritual of sharing the blood and flesh of Jesus remained, and it became an excuse to give up the ways of sharing, eating, being full, and becoming a member of the family.

As a result, the early church, which aimed to become a communal community, sacramentized the Lord's Supper, ultimately relegating it to a superficial event and concealing it in the abstract language of Koinonia, the community of love (Ahn 1993a, pp. 388–89). In addition, the church gradually shifted from a living community to a worshipping community, and in this process, Jesus became the Christ and the object of worship rather than a figure who was with *minjung* (Ahn 1993a, p. 199).

However, Ahn thought that the table community, which had failed in the early days, would rather be better developed by Koreans. Ahn (1993a, p. 390) wrote that "Koreans who call their family members *shikgu* [literally meaning "eating mouth"] are the people who can understand Jesus the best." As a basis for this, Ahn makes three arguments. First, he pays special attention to the Korean word *uri*, literally meaning 'we' or 'our.' According to Ahn (2019, p. 249), *uri* in Korean "means not the plurality of individuals but a community of shared destiny and is thus different in nuance from the Japanese word *wareware* or the English word 'we'." The Korean word for a pen made for keeping domesticated

animals also uses *uri* as a suffix, as in *souri* (cow pen) and *dwaejiuri* (pig pen), implying that "those in the same *uri* share the same destiny" (Ahn 2019, p. 249). In addition, another language habit of Koreans that reveals that the community is given priority over the individual is the predominant use of the pronoun "our" (*uri*) instead of the pronoun "my" in reference to many things. For example, Koreans are prone to say "our country" (*uri-nara*), "our school", and "our company" instead of "my country", "my school", and "my company", respectively. The same goes for references to more personal things and individuals, e.g., Koreans say "our house", "our wife", "our husband", and "our child" instead of "my house", "my wife", "my husband", and "my child", respectively.

Second, as insinuated above, Ahn pays attention to the fact that Koreans, who have internalized the priority of the community over the individual, use the word "*shikgu*" to refer to the family. *Shikgu* literally means "eating mouths", implying that family members share all the food. Ahn claims that the use of such a word is not the product of an idea created by a specific individual or group but is the result of the Korean people's experience of hunger and poverty caused by wars and natural disasters. In addition, Ahn (1993a, pp. 390–91) emphasizes that this word comes not only from the experience of starving people without a single grain of rice but also from their experience of being alienated and derided by the ruling class, although they are the principal agents of material production. Ahn thus discovers the possibility of an organic community that can reproduce the failed table community of the time of Jesus, based on the Korean experience implied by the use of such words as *uri* and *shikgu*. In fact, Ahn recreates the meaning of the table community through rituals for dead ancestors in Korea. He said that in Korea, memorial tablets are enshrined for deceased ancestors, and food is offered to their spirits at every meal, after which all family members eat it together.

> The important thing here is not just eating, but eating together with our ancestors. Sharing the food that was offered to our ancestors is what makes us a family (*shikgu*). That is why we use the expression 'a relationship to eat the rice from the same pot' to refer to close relationships. (Ahn 1993a, pp. 392–93)

Third, he dreams of realizing the table community through Korea's village ritual. Each year, villagers slaughter a cow or a pig, depending on the season, offer it as a sacrifice in a consecrated place, and then everyone equally shares the food that they cannot taste at any other time of the year. Sharing at this time is an event in which everyone experiences one another as *uri* with a joyful heart, transcending the difference between the rich and the poor (Ahn 2019, p. 251). Ahn refers to a story in the Bible as an event similar to the village festival in Korea: "Consider the people of Israel: Do not those who eat the sacrifices participate in the altar?" (1 Corinthians 10:18). Ahn also refers to the Korean poet Kim Chi-ha, a then-*minjung* poet and life activist, who wrote the following poem, which is in congruence with the former's thoughts:

> Rice is the sky.
> As you cannot have the sky to yourself
> You eat rice together with someone.
>
> Rice is the sky.
> As you see the stars on the sky together with someone
> You eat rice together with many others.
> When the rice goes into your mouth
> You enshrine the sky in your body.
> Rice is the sky.
> Ah, rice is
> What we eat together with all. (Ahn 2019, p. 251; Ahn 1997, p. 281)

Ahn thus tries to emphasize the realization of the sacramental life shown in Kim's poetry as a way to connect Christ with the life of the world and highlight a Christian life that celebrates life. What this means is that we share and eat together as a community. This

implies his wish for how, just as the church shares and eats together, this will expand into a social issue and society will become a community where people share (Ahn 1997, p. 281).

*2.3. Imagination of Ecological Thinking in Ahn's Theology*

Ecological thinking demands that we expand our sense of who we are and how we relate to one another and to the world (Ecoscenography 2023). Indeed, fundamental to ecological thinking is understanding how interrelated we are and how humans and the environment are interdependent. While Ahn did not mention such specifics about ecological thinking, he did try to reenact, through Koreans, the table community that Jesus thought about at the time of the early church, in which material equality and well-being were realized, transcending everything, regardless of class or status, wealthy or poor, men or women, and young or old. It is from Ahn's desire for a table community that we can imagine the ecological thinking of the people's community (*minjung* community) that he pursued. The community he wanted to create was an organic community in which the conflicting relationships of humans versus humans, humans versus nature, and nature versus nature were overcome and harmonized. It is based on Ahn's ecological thinking that an individual cannot live alone but that two or more people live together and create a new world through labor. Labor is the overall result of human actions that love nature and is a visible field of life phenomenon in which humans and animals, plants, and minerals engage in interrelated acts.

This ecologically oriented reasoning, as noted by Ahn (2019, pp. 246–47), is revealed by the three types of modes of existence classified by Pitirim A. Sorokin, a Russian-American sociologist: collective juxtaposition, indirect coexistence, and integral community (community of shared destiny). First, collective juxtaposition refers to a mode of existence that is like a collection of all sorts of items in a trash can. Trash bins contain a mixture of various items that are thrown away, such as papers, glass fragments, broken dishes, and plastics, as well as organic matters such as fish scraps, rotten fruits, and cookie crumbs. However, these things in the trash bin are not there together, or put together, out of necessity or because they help one another. They are in the same place because they are treated as useless, thereby coexisting by chance (Ahn 2019, p. 246). In human society as well, there are coexisting groups intertwined by coincidence and irrelevance, like junk in a trash can.

The second mode of existence is indirect coexistence. As in collective juxtaposition, there is no necessity for coexistence, although coexistence happens for an indirect reason. For example, a ballpoint pen, a notepad, a handkerchief, cosmetics, a comb, a mirror, and the like can coexist in a person's bag. The coexistence of the things in the bag has not occurred for a direct reason, for it is indirectly caused by the user's needs: "These various items are in an indirect relationship with one another according to the needs of the person who has them" (Ahn 2019, p. 246). Likewise, in today's human society, indirect coexistence, in which people gather around a specific place and then disperse due to capital, power, or personal interests, appears everywhere.

The third and last mode of existence is organic community. This mode of existence is not a coexisting relationship shaped by any external reason, forcibly or voluntarily or directly or indirectly, but is a community of shared destiny connected to a single network. Ahn (1993a, p. 387) describes this organic relationship as follows:

> This means that I cannot exist without you; your existence is unthinkable without me; your joy becomes my joy; my pain becomes your pain; the pain of one becomes the pain of all; and when a member is lost, he or she cannot be replaced like a machine part, for the pain from such a loss must simply be lived with.

Ahn (2019, p. 247) identifies this organic community with what Paul called the community of Christ, *sōma Christou* (the body of Christ). *Sōma* is a concept that opposes the worldview of spirit–body dualism marked by *sarx*, which corresponds to a lump of meat or flesh, and *pneuma*, which corresponds to spirit. *Sōma* thus refers to a body in which both are integrated into one (Ahn 2019, p. 247; Ahn 1993a, p. 387). As Paul used this word to overcome the dualistic worldview based on a flesh–spirit dichotomy, Ahn does the same:

"The body of Christ is constituted by Christians and intended to be the church as an organic community" (Ahn 2019, p. 247). The attempt to realize the community as the body of Christ was the form of the early church with a communal characteristic. This type of community takes the primitive Israeli social system as its prototype. First of all, primitive Israel acknowledged God's sovereignty over all things. Under God's sovereignty, Israelites autonomously solved all the problems of the community through direct democracy. The important fact is that they did not recognize private ownership of land but only recognized the community's right to enjoy it. The social system of primitive Israel thus refers to a system that developed into a form of communal participation and possession against power and privatization (Kang 1992, p. 76). Won-don Kang (1992, p. 76), a Christian ethicist, summarizes this social order based on the concept of publicness as follows:

> The struggle of free peasants against large land ownership, the prophetic movements against the king's despotism, bureaucratic corruption, militarism, and the tyranny of the wealthy, and various forms of liberation movements against foreign domination all have primitive Israel as their historical and embryological nucleus. It is no exaggeration to say that these are movements aimed at restoring public social order, and that Jesus' movement also shares the same axis in this respect.

However, the biggest culprit that destroys the organic community, the table community, is the privatization of publicness, which privatizes the public domain. As evidence for this, Ahn exemplifies the forbidden fruit of the Garden of Eden. In asking what the forbidden fruit is and why it was planted, he understands it as something that no one should privatize, monopolize, or make his or her own and that it belongs to the public:

> I first describe it as publicness. It is not certain if it is something that everyone can receive together, but something that should not be privatized is publicness. The sky is public. No one can privately own the sky. Earth was also originally public. You can cultivate it and share what it produces, but you cannot privatize it. (Ahn 1993b, p. 439)

Ahn considers all forms of privatization of publicness as "sins", including Anglo-Saxons plundering North America and Australia by monopolizing and privatizing the land, regal power privatizing power to monopolize the system of power, the privatization of human rights by the rulers, and the privatization of God to monopolize God by confining it to a specific religion (Ahn 1993b, pp. 440–42). Ahn thus interprets the lost paradise as a result of the privatization of publicness: "Privatization is the greatest enemy that destroys the organic community and makes the table community impossible" (Ahn 2019, p. 253). And the root of this public idea is found in the *minjung* tradition of the Bible, and the gist of it lies in the idea that the land is "God's", e.g., "The land must not be sold permanently, because the land is mine, and you reside in my land as foreigners and strangers" (Leviticus 25:23). According to Ahn, since the land belongs to God, no human being can claim permanent ownership of the land. The idea of the land as "God's property" means that while it is for everyone, it cannot belong to anyone (Ahn 1990a, pp. 205–6).

However, God's things, namely publicness, are not limited to the land, for all things, including nature and animals, are public. After God created all things (material things or the material world), he created humans as its companions. This implies that humans should never monopolize or privatize the material world, which means, by extension, that they are responsible for its wellbeing (Ahn 1990b). Also noteworthy is that Ahn's critical stance on privatization is consistent with the aforementioned viewpoint of green Christianity and that he would have been vehemently opposed to actual cases of privatization of public services, especially those involving water and electricity, in many parts of the world.

### 2.4. Publicness and God, and the Kingdom of God

According to Minjung theologian M.-s. Kim (2011, pp. 268–69), there are three factors that influenced Ahn's philosophy and theological thought: (1) the *minjung* incidents that

erupted in the process of modernization in Korea in the 1970s; (2) the theological trends of Western modernity; and (3) the spiritual culture of Korea. More specifically, the *minjung* incidents, the first factor, defined the underprivileged class of society as *minjung*, who played a leading role in economic development during the modernization era but were excluded from its benefits. The scene of the people's suffering and their struggle for survival greatly influenced Ahn's theological thought. The second factor, the Western modern theological trend, was formed when Ahn studied abroad in Germany. While studying at Heidelberg University, he was influenced by existential philosophers and theologians and used historical criticism and sociological methodology as theological tools to form his thoughts. The third factor, Korea's spiritual culture, was formed through his activities in Korea after his return from studying abroad. He introduced and disseminated Western theology in Korea, as well as critically assessing the main characteristics and limitations of Western theology. More importantly, Ahn was influenced by Korean and Eastern religious ideas, especially those of Yoo Young-mo (1890–1981), who was a Christian philosopher and educator, culminating in the formation of Ahn's original and distinctive theology (M.-s. Kim 2011, pp. 268–69).

First of all, Ahn was influenced by Yoo's various intellectual approaches, especially his view of God. Yoo's view of God can be condensed into two basic ideas: "God is like a father who loves his children (while the latter respects the former)" and "God exists as a non-being." The former was influenced by Confucianism, while the latter was influenced by Taoism. Yoo borrowed the Confucian idea of filial piety and defines Christianity as the religion of the father (*Abba*). Based on the universal love of humanity, he advocated the "theology of filial piety", which emphasizes the need for humanity to avoid selfishness or national egoism, serve God, the owner of heaven and earth, as our father, and obey his words (M.-s. Kim 2011, pp. 271, 276).

Also, Yoo, informed by Taoist thoughts, defines God as one "who exists as a non-being." According to Taoist philosophy, all things in the universe are living, dynamic beings, so the reality of all things in the universe cannot be objectified. Taoist philosophy defines these living, dynamic beings as "being without being" (無有之有), "existence that is non-existence" (不存之存), "shape without shape" (無狀之狀), and "form without form" (不形之形). Therefore, all things can be said to be structures that exist while being non-existent and that do not exist while being existent. That is, things in the universe that are non-existent have no fixed shape and cannot be said to exist or not exist (Won 1997, p. 32, cited from M.-s. Kim 2011, p. 277). Inspired by such a Taoist view, Yoo explains that God cannot be captured as an object of human sensory experience or reason and that God is "the one who possesses the dimensions of nothingness and emptiness and who encompasses the dimensions of existence." Yoo thus defines God as a being that is neither a being nor a non-being, i.e., one "who exists as a non-being'" (Park 2001, p. 86, cited from M.-s. Kim 2011, p. 276).

Influenced by Yoo's thoughts, Ahn opposes the objectification of God. Many traditional theists believe that God has five metaphysical personal attributes: simplicity, immutability, eternity, painlessness, and insensibility (Wainwright 1999, p. 34). However, Ahn denies this and criticizes the theistic thinking of modern theists as follows:

> He (God) is not someone who responds when humans ask questions about the riddle of the universe out of intellectual interest, nor is he the one who listens to and resolves the situation of human beings who are in agony as they suffer from corruption and conflict… Yahweh is not omnipotent, nor is he a goblin bat, nor does he appear as an immutable principle. (Ahn 1993a, p. 166)

The reason Ahn himself rejected the objectification or personification of God is because an objectified being is bound to be trapped and limited by the objectified image. God, conceptualized as a person or expressed in language, cannot be the original God. M.-s. Kim (2011, p. 230) explains the reason why Ahn refused to personify or objectify God as follows:

> What is language? Is it not a social promise? Therefore, language cannot help but be restricted by society. The thought system is a product of language. It can only be defined by language. The moment every object is expressed in language, it becomes trapped in the frame of language and loses its original appearance. All reality is transformed when expressed in language. Only when concepts are dismantled does reality appear.

Ahn's view of God is also manifested through his unique understanding of the Holy Spirit. He understands the Holy Spirit as "the energy of life, that is, *qi*, originating from God, and interprets it as an event rather than viewing it as a personal being based on the dichotomy of spirit and body" (Cha 2006, p. 72). This is because "*qi* is inherent in life and, at the same time, fills the entire universe as an intermediary energy that connects life to life" (Cha 2006, p. 72). In this way, Ahn's view of God, which does not conform to any image, resonates with the basic thoughts of the Diamond Sutra, which is a very popular Mahayana text in Northeast Asia and is best known for the 18 "Wisdom" texts and accompanying commentaries (the text is composed of dialogue between Buddha as teacher and a disciple as questioner). According to the Diamond Sutra, all things do not exist as fixed or isolated entities but exist in relation to one another. In other words, all objects exist in a dependent manner and exist as changes and processes. Ahn understands that God is not defined as a fixed frame, personality, or image but is experienced in life (M.-s. Kim 2011, p. 231).

Going one step further, Ahn recognizes God as publicness. In arguing that "God should also be considered publicness", Ahn (1993b, p. 439; 1997, p. 17) reasons that God cannot be owned by anyone and that no sect or religion can or does monopolize God. Also, to the question of "What is God?" Ahn (1993a, p. 396) answers that "He is publicness." Such an answer is based on Ahn's rejection of the Western Christian conception of God, who is regarded as a transcendental personality, and on his reinterpretation of Yoo's view of God, i.e., "God who exists as a non-being." The former is a resistance against the ideological view of God, while the latter represents a reinterpretation of the Eastern philosophical way of thinking. Based on this logic of thought, Ahn understands the Kingdom of God as publicness. He expresses the Kingdom of God, which "cannot be expressed in human words", in "present language" that people can understand (Choi 2006a, p. 209).

> What is the Kingdom of God, really? It is to turn publicness into publicness, i.e., it is not privatized. Restoring to publicness what has been divided and torn by privatizing everything, including politics and economy, is inseparable from the achievement of God's Kingdom. From the point of view of *minjung*, there is no need for a Kingdom of God that has been repeatedly subjected to mentalization, becoming otherworldly and conceptualized. (Ahn 1987, p. 246)

Ahn does not think of God and the Kingdom of God as dualistically separate. To him, God and the Kingdom of God are not separate and independent entities, but the way God exists is the Kingdom of God, and that is publicness. Therefore, Ahn does not understand God or the Kingdom of God as a product of ideation represented in indescribable metaphysical language but reduces it to something that can be made present and historicized. Hyung-muk Choi (2006a, p. 210) views this as "developing democratic systems and public ownership by returning materials and power to the original producers and further developing an ecological community where humans and nature coexist." Publicness is a "zone" of interconnected shared life phenomena that cannot be owned by anyone, i.e., the world is basically interconnected and humans are not separate from the rest of the universe. Therefore, publicness corresponds to an ecology that cannot be monopolized by anyone, which Ahn symbolizes as God and God's Kingdom. In short, for Ahn, God and God's Kingdom are ecology, and ecology is God and God's Kingdom.

### 2.5. Characteristics of Ahn Byung-Mu's Ecological Thought

Ahn Byung-Mu's ecological thought is best manifested in his view of God. What is unique about his view of God is not only his resistance to the traditional Western view of God based on dualism but also his reinterpretation of the Eastern philosophical view of

God centered around Yoo's thought. Regarding the former, Ahn insists that a God who is not reified is not God and emphasizes that God is not a personal entity that exists in the world beyond, i.e., in the other world and separate from the world of phenomena. In that respect, it is emphasized that love, blessings, and grace should be incarnated without being confined by abstract or sweet religious language. Regarding the latter, M.-s. Kim (2011, p. 279) explains as follows: Yoo's view of God—"God existing as a non-being"—which emphasizes the publicness of God—is not limited to simple philosophical principles, as it contains the practical dimension of "the practice of reaching emptiness through sharing and completing sharing through emptying, i.e., a life that practices sharing through emptying".

Ahn contrasts Yahweh and Baal in order to concretize the meaning of this view of God. First of all, the former God is a historical God, not an entity that settles in or occupies a certain space, but a God who continues to move forward by existing in time, that is, to aim or pursue. On the other hand, the latter god is the god of the earth, a god who is closely attached to the land and enriches it; that is, a god of possession. Accordingly, the community that has faith in the latter god is a community that places emphasis on possession and occupation and desires preservation and expansion. In contrast, the community that pursues the former god is a community that overcomes the old community occupied by the possession of space and moves through time; that is, a community that pursues historical orientation and aims for hope and promise. This new community can continue to exist because it seeks hope and promise and pursues a temporality oriented toward history rather than pursuing spatiality that will one day become outdated and obsolete. Ahn (1999, pp. 579–83) sees this new community that has escaped from spatial (possessive) constraints as the Kingdom of God. Therefore, Ahn (1999, p. 584) rejects a community that emphasizes self-guard and self-expansion by guaranteeing ownership; instead, he dreams of building a community that breaks down the wall of spatially vested interests by completely abandoning oneself.

As such, Ahn's view of God is a materialistic interpretation approach that implies the dimension of historical and sociological relationships and seeks to replace transcendence with phenomena, ideas with materialization, and spatiality with temporality. In other words, he emphasizes the reification of God and the Kingdom of God.[7] Furthermore, as Hyung-muk Choi (2006a, p. 210) points out convincingly, the realization of the Kingdom of God for Ahn means changing all social relationships based on exclusive privatization into public relationships. Here, the reification or incarnation of God is "sharing", which is the foundation of an organic community (Choi 2006a, p. 210).[8] Sharing is a visible course of reification and the foundation of the existence of an organic community. Therefore, God is none other than publicness; publicness is reification; reification is sharing; and sharing is the basis of an organic community, all of which leads to an ecology where God can be said to be the nucleus of an organic community.

This thought system implies the nature of criticism and resistance to what Jacques Derrida calls the "metaphysics of domination" of Western metaphysics that justifies exclusion and violence, based on the premise of the unity of thought and existence (N.-i. Kim 2007, p. 289). It goes against the universality centered on one and the exclusivity that seeks to converge on one. Therefore, ecology, as reinterpreted by Ahn, does not mean physical nature. He blatantly states that he is not interested in nature.

> I am not interested in nature itself. I have never experienced God in nature. Even if I read a novel, I lose interest in the depiction of nature and focus on the description of humans… Even though I live in nature day and night, I do not start from there and see people. One may experience God through nature or experience the state of mystery through something else, but I am not interested in such things. I only encounter God through events that occur between people, and I have no interest in anything else. (Ahn 1993a, p. 183)

Ahn understands ecology not from a natural perspective but as an "event" that implies a materialistic interpretation. This is because Ahn, to whom ecology is publicness and shared, reinterprets it thoroughly materialistically. However, this does not refer to

the practical implementation of the communist ideology of "work according to ability and distribute as necessary" (Noh 1992, p. 645) but refers to a community with religious significance premised on thorough self-sacrifice and autonomous spontaneity.

The event in which sharing, the basis of such an organic community, was realized in Korea, that is, the incarnation of publicness and ecology, happened on 13 November 1970, when Jeon Tae-il committed self-immolation. Jeon, a 22-year-old Christian at the time, was living as a factory worker whose human rights were severely violated. His monthly salary was 1500 won (or $4.74 at the exchange rate of 316.7 won per $1 in 1970), although the minimum daily meal cost was 120 won, amounting to about 13 percent of his monthly salary. This meant that even if he worked every day for a full month, he could not earn enough to cover the minimum cost of food. That is why he had to earn extra income by getting up early in the morning to shine shoes. He also sold chewing gum in the evening. In order to improve the miserable, subhuman lives of himself and his co-workers, Jeon prepared questionnaire data to expose the plight of the workers, showing them to employers and the government labor agency, but to no avail. He also appealed to pastors of large churches, but even this effort bore no fruit. As a poor worker who had nothing to lose, Jeon decided to offer himself as a living sacrifice and died by self-immolation. Ahn (1993a, p. 400) sympathized with Jeon Tae-il for this incident, saying, "It was not just an act of sharing his food with others, but was the act of sharing his body." Ahn (1993a, p. 400) also argued that Jeon's sacrifice was an act corresponding to Jesus' cry of "Eat my flesh. Drink my blood" (Ahn 1993a, p. 400).

In this way, for Ahn, sharing a meal, that is, an event in which sharing is materialized in reality, is an ecological event in which God becomes incarnated and publicness becomes a "thing" or "phenomenon." Such an ecological event becomes possible through *minjung*'s self-transcendence. Self-transcendence refers to "a new group, a group that is revived from despair and resignation, and a true *minjung* that transcends one's talent, one's character, one's helplessness, and one's potential" (Ahn 1993a, p. 411). Such a self-transcendent act is shown not only in the Jeon Tae-il incident but also in the prayer of a mother whose son was imprisoned for the Mincheong Hakryun Incident of 1974.[9]

> … please have pity on these people and this country. Please forgive me for my past thought that I would only feed and clothe my children well. Please forgive me for not understanding the true meaning of suffering on the cross and for only wanting my child to succeed in this world. Please forgive me for my sin of being reluctant to become a neighbor to my poor neighbors. Also, please forgive the sins of mothers who enjoy luxury while ignoring the suffering of this country's underprivileged widows and orphans… (Ahn 1993a, pp. 413–14)

Ahn does not stop here. "Sharing", the basis of the organic community planned by Ahn, seems to operate not only at the social and historical level but also in the realm of the invisible microscopic world. In 1988, Ahn founded a magazine entitled *Salim*, literally meaning "living", as he was influenced by the idea of "universe life" of C.-h. Kim (1996). According to Kim Chi-ha, life includes not only organisms capable of self-replication but also matter. This is because life is not an entity but a creation, and therefore, life as a creation does not stay constant even for a moment and continues to change in its relationship with everything. He focuses on the fact that life has a center and a marginal space surrounding it, and he pays special attention to the "hidden order" in which other living beings obtain food by engaging in this marginal space.

Based on this, Ahn defines *salim* as cherishing life in the hidden order, nurturing it according to its individuality, and further realizing it in full bloom (C.-h. Kim 1995, 1996; cited from N.-i. Kim 2007, p. 274). The mechanism of life phenomena in the microscopic world that Ahn seeks to realize through *salim* is an ecological worldview that encompasses and penetrates the center and the margins and operates through mutual interaction between the two equal entities (the center and the margin). With this intuition about life phenomena in the microscopic world, Ahn took a greater interest in nature than in *minjung* in his later years and referred to the life community of the forest, where life forms in

the forest depend on and relate to one another and form one huge web of life, a community (*Gemeinschaft*), not a society (*Gesellschaft*) (M.-s. Kim 2011, pp. 236–37).[10]

In short, the ecological ideas hidden in Ahn's *minjung* community are, first, an ecological event in which publicness is reified and sharing is incarnated through self-transcendence,[11] and second, they can be said to be a characteristic that aims for the phenomenon of life-saving in which the center and the margin are viewed as equals and interact with each other. From this, it can be seen that Ahn's ecological thought has an ambivalent aspect of materialistic resistance against the philosophy of Western metaphysics by materializing ideas while at the same time adding the spiritual and mystical characteristics of Eastern philosophy, including those of Lao-tzu and Yu Young-mo. Would this not be the *minjung* community that Ahn was aiming for? We envision the *minjung* community that Jesus attempted but failed to realize among the *minjung* of this land living as subalterns today by borrowing Ahn's theological imagination to uncover hidden ecological ideas.

## 3. Conclusions

Ecological thinking demands that we expand our sense of who we are and how we relate to one another and to the world (Ecoscenography 2023). Fundamental to ecological thinking is understanding the basics of ecology and living systems. Ecology shows how ecosystems are more than merely a collection of species; they are also relational systems in which humans are connected to other biological systems like animals and plants (Ecoscenography 2023). As a whole, ecology promotes a deeper knowledge of how interrelated and interdependent the environment is. In this paper, we were able to gain a glimpse of Ahn Byung-mu's ecological thought by discovering the characteristics of his ecological ideas that seemed to be hidden in his conception of the *minjung* community. In his discussion of God, Schleiermacher (1997, p. 117) said that "Imagination is the highest and most fundamental thing among humans, and everything else is merely a reflection of it." Imagination is not an arbitrary manipulation of human consciousness but rather a human representational ability that objectifies the world outside of humans. In that context, we were able to infer the hidden ecological thoughts of Ahn through our interpretation of his imagination of community.

In a word, the basic reason or motive behind the organic community that Ahn dreamed of is ecology. However, since acts that go against the organic community are a privatization of publicness, it can be said that Ahn's basic ecological position is based on his concept of *gong*. Also, it is revealed that publicness is not irrelevant to Ahn's view of God, that publicness is God, and that the place where publicness is realized in reality is the Kingdom of God. Based on the above observation, we can infer the ecological reasons hidden behind Ahn's imagination of community as follows: First, in the macroscopic and visible world, Ahn's ecological thought is manifested in the idea that sharing is an event that is materialized in social and historical reality. In the microscopic and invisible world, moreover, the life phenomenon of the living is expressed through the interaction between the center and the margin, encompassing the organic and material domains. Second, Ahn's ecological thought is not a utopia that does not exist in this world, but a kind of Christian movement based on the religious imagination of his dreams. Third, such ecological imagination is not a tool of efficiency bent on linear and internalized, complacent optimism but a practical driving force for reforming reality laid out along a clear path. At first glance, the ecological thought drawn by Ahn may appear to be a broken, fragmented, and fragile representation, but it is an "event" in which imagination transforms into reality, spatiality into temporality, and ideas into objects or reality.

In that respect, the ecological community conceived by Ahn is not submerged thoughts buried in the past, but a part of the event theology rising like a volcano here today. Therefore, we can borrow the imagination of Ahn's ecological thinking and reserve the gloomy judgment that regards Minjung theology as a theology of the past in an era without *minjung*. Instead, we anticipate that when Minjung theology is accepted as a vibrant event at all times and everywhere, the ecological community envisioned by Ahn will be able to be

newly regenerated in any era and anywhere. As can be seen from Mother Teresa's claim that "there is no peace without sharing", it shows that ecological soundness, peace, and coexistence can be achieved when there is a desire to share on earth (G. Lee 2010, p. 21). Therefore, Ahn's organic ecological community can be said to be not an abstract thought but a realistic embodiment of a vision becoming reality.

**Author Contributions:** Conceptualization and writing original draft, J.K.; writing second draft and editing, A.E.K. All authors have read and agreed to the published version of the manuscript.

**Funding:** This research was funded by the Ministry of Education of the Republic of Korea and the National Research Foundation of Korea [NRF-2021S1A5C2A02088321] and a Korea University Grant.

**Institutional Review Board Statement:** Not applicable.

**Informed Consent Statement:** Not applicable.

**Data Availability Statement:** No data are available, for no data were used for the article and no new data were created.

**Conflicts of Interest:** The authors declare no conflict of interest.

## Notes

[1] While Christian denominations differ on their views on ecological issues, conservative Christians, especially members of the Christian Right in the United States, stand out because they are generally less concerned about environmental issues than the general public. Some fundamentalist Christians even deny that climate change and global warming are happening.

[2] Minjung theology is a form of liberation theology that emerged from the experience and perspective of South Koreans in the 1970s, who at the time were struggling against economic injustice and dictatorship (A. E. Kim 2018; Y. B. Kim 1983). The first generation of leading minjung theologians includes Ahn Byung-Mu, Suh Nam-dong, Hyun Young-hak, Moon Dong-hwan, Han Wan-sang, Kim Yong-bok, Byungsung Huh, and Suh David Kwang-sun, all of whom were active during the democratization movement of the 1970s (J. Kim 2022). Among them, two central figures in Minjung theology are Ahn Byung-Mu and Suh Nam-dong. The second generation of Minjung theologians was particularly active in the 1980s and included such scholars as Park Sungjun, Kang Won-don, Park Jaesoon, and Kwon Jin-gwan. The third generation of Minjung theologians, who have been active since the 1990s, include Kim Gyungho, Choi Hyung-muk, Kim Jin-Ho, and Kim Myung-su.

[3] It is also worth noting the ecological views of Suh Nam-dong, another prominent scholar of Minjug theology on the same footing as Ahn. Suh (1976) delved into ecological theology as early as the mid-1970s, writing about the importance of striving for ecological ethics in a chapter entitled "Towards Ecological Ethics" in his book entitled *Jeonhwansidaeui sinhak* (Theology in the Age of Transition), published in 1976 (see Suh 1983; H. Kim 2013; Kang 2006).

[4] Ahn was born into a family that had been in the oriental medicine business for generations. Ahn's grandfather was also a doctor of oriental medicine and a scholar of Chinese classics who subscribed to a strict patriarchal worldview. Ahn recounted that when his grandfather sometimes brought a woman into the house, his grandmother politely greeted them both and prepared bedding. Due to the influence of such a family background, Ahn's father apparently also conducted himself in the same way, and Ahn is said to have developed resentment toward his father's behavior throughout his life. For example, he hated his father so much that he spent his entire life trying not to resemble the latter's handwriting style (N.-i. Kim 2007, pp. 28–29; Ahn 1996, p. 167).

[5] *Qi*, typically translated as "vital energy" or simply "energy", is considered to be a life force forming part of and permeating every living entity (https://en.wikipedia.org/wiki/Qi, accessed on 29 November 2023). There is thus no life form that is not infused with *qi*. Qi has been an important concept in many Chinese and Korean philosophies and is the central underlying principle in oriental medicine. In traditional Chinese medicine, for example, the flow of *qi* must be unhindered for health and longevity. It is also believed that comprehending the flow and rhythm of *qi* could guide exercises and treatments for the body to facilitate stability, health, and longevity (https://en.wikipedia.org/wiki/Qi, accessed on 29 November 2023).

[6] Hyanglinwon was used as a famous high-class restaurant called Hanatsugi (花月) run by the Japanese during the Japanese colonial period. After the liberation in 1945, it was operated as an orphanage under the name of Hyanglinwon, although the North Korean army also used it as a medical unit during the Korean War. Hyanglinwon was equipped with a large Korean-style house surrounded by six detached smaller buildings and even a pavilion on a spacious site of nearly 4300 m$^2$. Ahn was able to use Hyanglinwon, which was one of many buildings and lands that were sold to Koreans by the Korean government following the departure of the Japanese owners after the liberation, with the permission of an elder of a Methodist church, who had the possessory right to the building. Ahn moved in first with his mother, repaired the houses, and then Ilsinhoe friends joined in and started living together (N.-i. Kim 2007, pp. 92–93; see Hyanglin Church website, http://www.hyanglin.org/bbs/52809, accessed on 21 November 2023).

7  A working definition of "reification" for the paper is "the process of transforming an abstract concept or thought into something concrete", without any Marxist slant.

8  Ahn (1993a, p. 398) says that truth and grace become reality only when they are shared, thereby forming a new community.

9  The incident refers to a case in which 180 individuals were arrested and prosecuted under a fabricated charge of attempting a communist people's revolution. The year 1974 was when the anti-dictatorship and democratization movements were gaining momentum against the then-president Park Chung-hee, who amended the so-called Yushin Constitution, granting enormous powers to the president, including no limits on reelection. Those who were charged in the Mincheong Hakryun Incident were largely students, many of whom were subjected to torture.

10  Elaborated by the German sociologist Ferdinand Tönnies, *Gemeinschaft* is associated with traditional and small-scale societies in which there exist intimate social relationships and strong group solidarity, while *Gesellschaft* is typified by a modern society in which social relationships are based on rational self-interest and impersonal, calculating acts.

11  An ecological event in which the public becomes reified and sharing becomes incarnated is possible when it is based on temporality rather than spatiality.

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
