# Peer review of "Minjung Theology of Korea and Ecological Thinking: Focusing on the Theological Imagination of Ahn Byung-Mu"

_religions, doi:10.3390/rel14121533_

Round 1

Reviewer 1 Report

Comments and Suggestions for Authors

Author Response

Response to Reviewer 1’s Comments

This paper is an attempt to discover the ecological thoughts of a Korean Minjung theologian by focusing on his theological imagination. While I agree with the overall argument, I would like to add a few points. Ahn Myung-Moo is not the only Korean Minjung theologian, but there are others such as Suh Nam-dong and those who succeeded him. Without mentioning these currents of Minjung theology, as well as Ahn’s theology, English-speaking readers may be misled into thinking that Ahn is the only Korean Minjung theologian. In this regard, a brief mention of the history and research of Minjung theology seems necessary.

  • Response: Footnote #2 has been greatly expanded to include more information about the leading scholars of Minjung theology, identifying not only the first generation minjung theologians but also second- and third generation minjung theologians.

The introduction is rather long compared to the overall discussion. It would be better to shorten the introduction and discuss only the key points related to Ahn’s ecological thoughts.

  • Response: Discussions on environmentalism of Buddhism, Hinduism, and Islam have been shortened into just one paragraph. Discussion on ecological liberation theology has also been shortened.

The core of this paper is to discover the sprouts or germs of Ahn’s ecological thoughts based on Ahn’s concept of God as Publicness and Yoo Young-Mo’s concept of God’s absence. The concept of God as publicness is known to have been presented by Ahn as an alternative to overcome economic inequality in capitalist society, as in the case of liberation theology. It is questionable whether we can find traces of Ahn’s ecological thoughts here. It would be good to have a more convincing explanation for this. Also, Yoo Young-Mo’s concept of God without God seems to have the potential to generate more ecological thinking than the former in terms of integrating the religious spirit of the East and the West. A more in-depth comparison of Ahn Byung-Moo’s concept of God in relation to Yoo’s concept of God would help clarify the thesis of this paper. It seems that this needs to be supplemented.

  • Response: We’ve added the following paragraphs on pp. 11-12.

      According to Minjung theologian Kim Myung-su (2011, pp. 268-269), there are three factors which influenced Ahn’s philosophy and theological thought: 1) the minjung incidents that erupted in the process of modernization in Korea in the 1970s; 2) the theological trends of Western modernity; and 3) the spiritual culture of Korea. More specifically, the minjung incidents, the first factor, defined the underprivileged class of society as minjung, who played a leading role in the economic development during the modernization era but were excluded from its benefits. The scene of the people’s suffering and their struggle for survival greatly influenced Ahn’s theological thought. The second factor, the Western modern theological trend, was formed when Ahn studied abroad in Germany. While studying at Heidelberg University, he was influenced by existential philosophers and theologians, and used historical criticism and sociological methodology as theological tools to form his thoughts. The third factor, Korea’s spiritual culture, was formed through his activities in Korea after his return from studying abroad. He introduced and disseminated Western theology in Korea as well as critically assessing the main characteristics and limitations of Western theology. More importantly, Ahn was influenced by Korean and Eastern religious ideas, especially those of Yoo Young-mo (1890-1981) who was Christian philosopher and educator, culminating in the formation of Ahn’s original and distinctive theology (Kim, M. 2011, pp. 268-269).

       First of all, Ahn Byeong-mu was influenced by Yoo’s various intellectual approaches, especially his view of God. Yoo’s view of God can be condensed into two basic ideas: “God is like father who loves his children (while the latter respect the former)” and “God who exists as non-being.” The former was influenced by Confucianism, while the latter was influenced by Taoism. Yoo borrowed the Confucian idea of filial piety and defines Christianity as the religion of the father (Abba). Based on the universal love of humanity, he advocated the “theology of filial piety,” which emphasizes the need to “break away from family selfishness or national egoism, serve God, the owner of heaven and earth, as our father, and obey his words” (Kim, M. 2011, pp. 271, 276).

       Also, Yoo, informed by Taoist thoughts, defines God as “the one who exists as Non-Being.” According to Taoist philosophy, all things in the universe are living, dynamic beings, so the reality of all things in the universe cannot be objectified. Taoist philosophy defines these living, dynamic beings as “being without being” (無有之有), “existence that is non-existence” (不存之存), “shape without shape” (無狀之狀), and “form without form” (不形之形). Therefore, all things can be said to be structures that exist while being non-existent and that do not exist while being existent. That is, things in the universe are “existences that are non-existences” that have no fixed shape and cannot be said to exist or not exist (Won 1997, p. 32, cited from Kim, M. 2011, p. 277). Inspired by such Taoist view, Yoo explains that God cannot be captured as an object of human sensory experience or reason and that God is “the one who possesses the dimensions of nothingness and emptiness and who encompasses the idea of ‘the One’ which contains the dimension of existence.” Yoo thus defines God as a being that is neither a being nor a non-being, i.e., a ‘being who exists as non-being’” (Park 2001, p. 86, cited from Kim, M. 2011, p. 276).

       Influenced by Yoo’s thoughts, Ahn opposes the objectification of God. Many traditional theists believe that God has five metaphysical personal attributes: simplicity, immutability, eternity, painlessness, and insensibility (Wainwright 1999, p. 34). However, Ahn denies this and criticizes the theistic thinking of modern theists as follows:

       He (God) is not someone who responds when humans ask questions about           the riddle of the universe out of intellectual interest, nor is he the one who           listens to and resolves the situation of human beings who are in agony as             they suffer from corruption and conflict…..Yahweh is not omnipotent nor is           he a goblin bat, nor does he appear as an immutable principle (Ahn 1993a,           p. 166).

       The reason Ahn himself rejected the objectification or personality of God is because an objectified being is bound to be trapped and limited by the objectified image. God conceptualized as a person or expressed in language cannot be the original God. Kim Myung-su (2011, p. 230) explains the reason why Ahn refused to personify or objectify God as follows:

       What is language? Is it not a social promise? Therefore, language cannot               help but be restricted by society. The thought system is a product of                     language. It can only be defined by language. The moment every object is           expressed in language, it becomes trapped in the frame of language and             loses its original appearance. All reality is transformed when expressed in             language. Only when concepts are dismantled does reality appear.

Reviewer 2 Report

Comments and Suggestions for Authors

I would recommend doing a bit more work on specifically Christian environmental activism on the first three pages; the list of: Hinduism says this, Buddhism says this, Islam says this... is a bit misleading because the heart of the paper is Minjung Christian liberation theology.

The portion on Bapsanggongdongche on pp. 6-8 is very interesting. I like how Ahn said that Bapsanggongdongche devolved into a "mere" holy communion. Usually Christian theologians would say that a sacrament is the highest form of holiness a material thing could be, but Ahn has it below table community, evidently. 

The privatization you describe on pp. 10-11 seems more important than some other topics that receive much more room. Privatization of water, for instance, is a scandal that Ahn would have been furious about!

The summary on the top of p. 15 is so valuable that it really should be previewed earlier in the article. And I think the conflict you describe there between Ahn and Western-style metaphysics is very interesting, but not really discussed enough in the article. 

Comments on the Quality of English Language

There are a few places where the definite article is used unnecessarily (a frequent issue for native speakers of Asian languages writing in English. 

The alternation of "I" and "We" is sometimes a bit confusing, unless there is some rhetorical strategy at work.

But nothing in the writing gets in the way of the communication of key ideas. 

Author Response

Responses to Reviewer 2’s Comments

Comments and Suggestions for Authors

I would recommend doing a bit more work on specifically Christian environmental activism on the first three pages; the list of: Hinduism says this, Buddhism says this, Islam says this... is a bit misleading because the heart of the paper is Minjung Christian liberation theology.

  • Response: I’ve added more remarks on Christian environmental activism, while deleting certain sections on Buddhist, Hindu and Islamic environmentalism.

The privatization you describe on pp. 10-11 seems more important than some other topics that receive much more room. Privatization of water, for instance, is a scandal that Ahn would have been furious about!

  • Response: We agree with the reviewer that this point is important, but we cannot really devote more space to it, because we’ve covered all that we could as far as what Ahn wrote about the issue. Nonetheless, we have added a sentence on p. 11, commenting that “Ahn’s critical stance on privatization is consistent with the aforementioned viewpoint of green Christianity and that he would have been vehemently opposed to actual cases of privatization of public services, especially those involving water and electricity, in many parts of the world.”

The summary on the top of p. 15 is so valuable that it really should be previewed earlier in the article. And I think the conflict you describe there between Ahn and Western-style metaphysics is very interesting, but not really discussed enough in the article. 

  • Response: We added the following in the first paragraph of p. 1:

     Due to such efforts, there is an increasing awareness that we must broaden           our understanding of our relationships to the planet in order to think                   ecologically. Realizing how we are interconnected to, and dependent on, the         environment is at the core of ecological thinking. Ecology demonstrates how       ecosystems are relational systems in which humans are interconnected with         other biological systems, such as plants and animals.

  • Regarding the latter point, we agree with the reviewer that it is an interesting point, but the conflict between Ahn and Western-style metaphysics lies outside principal focus of the paper. We believe that our discussion on the issue on pp. 12-13 would suffice, especially in light of the fact that the paper is rather long as is.

Comments on the Quality of English Language

There are a few places where the definite article is used unnecessarily (a frequent issue for native speakers of Asian languages writing in English. 

  • Response: We thoroughly went through the text and deleted unnecessary uses of the definite article.

“The alternation of “I” and “We” is sometimes a bit confusing, unless there is some rhetorical strategy at work.”

  • Response: Unlike the original version, i.e., the version that was uploaded onto Religions homepage, the version that was sent to reviewers did not indent all the long quotes. That is why many quotes were read as if they were remarks by the authors. All of this has been fixed now.